# Learning Local Search Heuristics for Boolean Satisfiability

**Emre Yolcu**
Carnegie Mellon University
eyolcu@cs.cmu.edu

**Barnabás Póczos**
Carnegie Mellon University
bapoczos@cs.cmu.edu

## Abstract

We present an approach to learn SAT solver heuristics from scratch through deep reinforcement learning with a curriculum. In particular, we incorporate a graph neural network in a stochastic local search algorithm to act as the variable selection heuristic. We consider Boolean satisfiability problems from different classes and learn specialized heuristics for each class. Although we do not aim to compete with the state-of-the-art SAT solvers in run time, we demonstrate that the learned heuristics allow us to find satisfying assignments in fewer steps compared to a generic heuristic, and we provide analysis of our results through experiments.

## 1   Introduction

Recently there has been a surge of interest in applying machine learning to combinatorial optimization [7, 24, 32, 27, 9]. The problems of interest are often NP-complete and traditional methods efficient in practice usually rely on heuristics or produce approximate solutions. These heuristics are commonly manually-designed, requiring significant insight into the problem. Similar to the way that the recent developments in deep learning have transformed research in computer vision [28] and artificial intelligence [43] by moving from engineered methods to ones that are learned from data and experience, the expectation is that it will lead to advancements in search and optimization algorithms as well. Interest in this line of work has been fueled by the developments in neural networks that operate on graphs [5] since many combinatorial problems can be naturally represented using graphs.

A problem that is becoming a popular target for machine learning is satisfiability [11]. Boolean satisfiability (abbreviated SAT) is the decision problem of determining whether there exists a satisfying assignment for a given Boolean formula. The task of finding such assignments if they exist or proving unsatisfiability otherwise is referred to as *SAT solving*. SAT is the canonical NP-complete problem [13]. It is heavily researched, and there exist efficient heuristic algorithms that can solve problems involving millions of variables and clauses. In addition to its fundamental place in complexity theory, SAT is practically relevant, and there is a plethora of problems arising from artificial intelligence, circuit design, planning, and automated theorem proving that can be reduced to SAT [37].

Assuming we are given instances from a known class of SAT problems, it is an interesting question whether we can discover a heuristic from scratch specialized to that class which improves upon a generic one. While handcrafting such heuristics for every single class is not feasible, an alternative is to sample problems from a class and *learn* a heuristic by training to solve the problems. In practice, we are often interested in solving similar problems coming from the same distribution, which makes this setting worth studying. In this paper we focus on stochastic local search (SLS) and propose a learnable algorithm with a variable selection heuristic computed by a graph neural network. Through reinforcement learning, we train from scratch a suite of solvers with heuristics specialized to different classes, and demonstrate that this approach can lead to algorithms that require fewer, although costlier, steps to arrive at a solution compared to a generic heuristic.

## 2  Related work

Lately, neural networks have seen an increased interest in applications to SAT solving. Selsam et al. [42] propose an approach where a graph neural network (called NeuroSAT) is trained to classify SAT problems as satisfiable or unsatisfiable. They provide evidence that as a result of training to predict satisfiability on their specifically crafted dataset, a local search algorithm is synthesized, through which they can often decode the satisfying assignments. In their work, the graph neural network itself acts as a learnable solver while we use it as a relatively cheap heuristic in an explicitly specified search algorithm. It is unclear whether their approach can be applied to learning distribution-specific algorithms as it is hypothesized to be crucial that NeuroSAT be trained on "patternless" problems, hence the reason for our approach. In a more recent work, Selsam and Bjørner [41] also use a simplified NeuroSAT to guide the search process of an existing solver. Another similar approach is that of Lederman et al. [31], where they attempt to learn improved heuristics to solve quantified Boolean formulas through reinforcement learning while using a heuristic computed by a graph neural network. A critical difference of our work from theirs is that we have a minimal set of features (one-hot encodings of the assignments) just enough to obtain a lossless encoding of the solver state while they make use of a large set of handcrafted features. On another front, Amizadeh et al. [1] use a graph neural network and a training procedure mimicking reinforcement learning to directly produce solutions to the Circuit-SAT problem.

SAT solving community has also experimented with more practical applications of machine learning to SAT, possibly the most successful example being SATzilla [48]. Closest to our approach are the works of Fukunaga [18, 19], KhudaBukhsh et al. [25], Illetskova et al. [22]. Briefly summarized, they evolve heuristics through genetic algorithms by combining existing primitives, with the latter two aiming to specialize the created heuristics to particular problem classes. We adopt a similar approach, with the crucial difference that our approach involves learning heuristics from experience as opposed to combining designed ones. There have also been other approaches utilizing reinforcement learning to discover variable selection heuristics [33, 34, 35, 29, 17], although they focus mostly on crafting reward functions. We turn our attention to learning with little prior knowledge, and use a terminal reward which is nonzero only when a satisfying assignment is found.

## 3  Background

### 3.1  Boolean formulas

We limit our attention to Boolean formulas in conjunctive normal form (CNF), which consist of the following components: a list of $n$ Boolean variables $(x_1, \ldots, x_n)$, and a list of $m$ clauses $(c_1, \ldots, c_m)$ where each clause $c_j$ is a disjunction ($\vee$) of literals (a literal refers to different polarities of a Boolean variable: $x_i$ and $\neg x_i$). The CNF formula is the conjunction ($\wedge$) of all clauses. Throughout the paper, $\phi : \{0, 1\}^n \to \{0, 1\}$ refers to a Boolean formula, $X \in \{0, 1\}^n$ to a particular truth assignment to the list of variables $(x_1, \ldots, x_n)$, and $\phi(X)$ is simply the truth value of the Boolean formula evaluated at assignment $X$. Also, $n$ and $m$ always refer to the number of variables and clauses, respectively.

### 3.2  Local search

SAT solvers based on stochastic local search start with a random initial candidate solution and iteratively refine it by moving to neighboring solutions until arriving at a satisfying assignment. Neighboring solutions differ by the assignment of a single variable, that is, the assigned value of a variable is *flipped* between solutions. Algorithm 1 shows the pseudocode for a generic stochastic local search algorithm for SAT.

As an example heuristic, the SelectVariable function of WalkSAT [39] first randomly selects a clause unsatisfied by the current assignment and flips the variable within that clause which, when flipped, would result in the fewest number

---

**Algorithm 1** Local search for SAT

**Input:** Boolean formula $\phi$, maximum number of trials $K$, maximum number of flips $L$

```
1:  for i ← 1 to K do
2:      X ← Initialize(φ)
3:      for j ← 1 to L do
4:          if φ(X) = 1 then
5:              return X
6:          else
7:              index ← SelectVariable(φ, X)
8:              X ← Flip(X, index)
9:          end if
10:     end for
11: end for
12: return unsolved
```

of previously satisfied clauses becoming unsatisfied, or with some probability (referred to as *walk probability*) it selects a variable randomly within the clause.

Our algorithm fits into the same template, with the difference being that the function SelectVariable employs a graph neural network to select a variable. Unlike WalkSAT we do not limit the selection to the variables in unsatisfied clauses, so it is possible to pick a variable that occurs only in currently satisfied clauses. However, similar to WalkSAT, with some probability we also randomly select a variable from a randomly selected unsatisfied clause.

### 3.3 Graph neural networks

Graph neural networks (GNNs) are a family of neural network architectures that operate on graphs with their computation structured accordingly [21, 38, 20, 5]. In order to explain our specific architecture in Section 4 more easily, here we describe a formalism for GNNs similar to the message-passing framework of Gilmer et al. [20].

Assume we have an undirected graph $G = (V, E)$ where $V$ is the set of vertices (nodes) and $E \subseteq V \times V$ is the set of edges. Further suppose that for each node $v \in V$ we have a row vector $h_v^0 \in \mathbb{R}^{d_V^0}$ of node features (possibly derived from node labels) with some dimension $d_V^0$. Similarly, for an edge $vw \in E$ between two nodes $v, w \in V$ we have a row vector $h_{vw} \in \mathbb{R}^{d_E}$ of edge features with some dimension $d_E$. A GNN maps each node to a vector space embedding by iteratively updating the representation of the node based on its neighbors. In this formalism we do not extract edge features. At each iteration $t \in \{1, \ldots, T\}$, for each $v \in V$ we update its previous representation $h_v^{t-1}$ to $h_v^t \in \mathbb{R}^{d_V^t}$ as

$$h_v^t = U^t \left( h_v^{t-1}, \sum_{w \in N(v)} M^t \left( h_v^{t-1}, h_w^{t-1}, h_{vw} \right) \right),$$

where $N(v)$ denotes the set of neighbors of node $v$. Message functions $M^t$ and update functions $U^t$ are differentiable functions whose parameters are learned. After $T$ iterations, we obtain the extracted features $h_v^T$ for each node $v$ in the graph.

For a compact notation, define the stacked node features $H_V^t \in \mathbb{R}^{|V| \times d_V^t}$ such that $H_V^t(i, \cdot) = h_{V(i)}^t$ where $V(i)$ is the $i$-th node in the graph under some indexing. Similarly define $H_E \in \mathbb{R}^{|E| \times d_E}$ to be the stacked edge features. Then a GNN with $T$ iterations computes $H_V^T = f_\theta \left( H_V^0, H_E, G \right)$ where $f_\theta$ is a function parameterized by $\theta$ that encodes each node of the graph $G$ as a real-valued vector.

## 4 Model

In this section we first describe the graphical representation for CNF formulas, then explain the exact input of the model, and finally define the architecture of the graph neural network that we use.

### 4.1 Factor graph of CNF formulas

We opt for a *factor graph* representation of CNF formulas. With this representation we obtain an undirected bipartite graph with two node types (variable and clause) and two edge types (positive and negative polarities). For each variable in the formula we have a node and for each clause we have a node of a different type. Between a clause and each variable that occurs in it there is an edge whose type depends on the polarity of the variable in the clause.

Note that unlike the graphical representation employed by Selsam et al. [42] and Lederman et al. [31], each variable (as opposed to a literal) has a corresponding node, which results in a slightly more compact mapping of a CNF formula to a graph. Figure 1 displays the factor graph of a simple CNF formula.

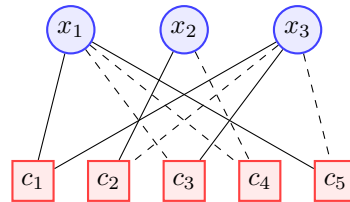

Figure 1: Factor graph of the formula $(x_1 \vee x_3) \wedge (x_2 \vee \neg x_3) \wedge (\neg x_1 \vee x_3) \wedge (\neg x_1 \vee \neg x_2) \wedge (x_1 \vee \neg x_3)$. Solid and dashed edges correspond respectively to positive and negative polarities of variables in clauses.

## 4.2 Input representation

For variable selection, we use a GNN that takes as input a formula $\phi$ with an assignment $X$ to its variables and outputs a vector of probabilities corresponding to variables (described further in the next section). The actual input to the model consists of the adjacency information of the factor graph and node features. Edge features are apparent from the adjacency matrices and do not act as explicit inputs to the model.

Since the factor graph of a CNF formula is bipartite and there are edges of two different types, we store a pair of $n \times m$ biadjacency matrices $\mathbf{A} = (A_+, A_-)$ such that $A_+(i, j) = \mathbf{1}\{x_i \in c_j\}$ and $A_-(i, j) = \mathbf{1}\{\neg x_i \in c_j\}$.

As node features we use one-hot vectors. When variable assignments are taken into account, there are three node types: variable assigned True, variable assigned False, and clause. More concretely, node features are stored as a pair $\mathbf{H}^0 = (H_v^0, H_c^0)$ of stacked variable features $H_v^0 \in \mathbb{R}^{n \times 3}$ and clause features $H_c^0 \in \mathbb{R}^{m \times 3}$. As a result, the pair $(\mathbf{A}, \mathbf{H}^0)$ is all that is needed to perform local search on a formula using the GNN variable selection heuristic.

For a single run of the local search algorithm, $H_v^0$ is set at first to reflect a random initial assignment to variables and at each search iteration the row corresponding to the variable selected by the heuristic is modified to flip its assignment.

## 4.3 Policy network architecture

In Section 3.3 we explained the abstract GNN architecture. The specific GNN that we implement, which we call the *policy network* along with the output layer, can be described as an instance of this architecture. In particular, it has four different message functions, one for each combination of nodes (variable, clause) and edges (positive, negative), and two different update functions for variables and clauses. The policy network runs for $T$ iterations, and as its components we have the following learnable functions, with dependence on parameters omitted for brevity, where $t \in \{1, \ldots, T\}$:

- $M_{v+}^t : \mathbb{R}^{d_{t-1}} \to \mathbb{R}^{d_t}$ and $M_{v-}^t : \mathbb{R}^{d_{t-1}} \to \mathbb{R}^{d_t}$ compute the incoming messages to each variable from clauses they positively and negatively occur in.
- $M_{c+}^t : \mathbb{R}^{d_{t-1}} \to \mathbb{R}^{d_t}$ and $M_{c-}^t : \mathbb{R}^{d_{t-1}} \to \mathbb{R}^{d_t}$ compute the incoming messages to each clause from variables that occur positively and negatively in them.
- $U_v^t : \mathbb{R}^{d_{t-1}} \times \mathbb{R}^{d_t} \to \mathbb{R}^{d_t}$ and $U_c^t : \mathbb{R}^{d_{t-1}} \times \mathbb{R}^{d_t} \to \mathbb{R}^{d_t}$ update the representations of variables and clauses based on their previous representation and the sum of the incoming messages.
- $Z : \mathbb{R}^{d_T} \to \mathbb{R}$ produces a score given the extracted node features of a variable.

In the actual implementation we have $d_t = d$ for $t > 0$, that is, the same at each iteration, and $d_0 = 3$ for input features. With a slight abuse of notation we will assume that we can apply the functions above to matrices, in which context they mean row-wise application. Having described the individual components, let $f^t$ be the function, with dependence on parameters omitted again, that computes the node representations of a graph with adjacency $\mathbf{A}$ at iteration $t$. We can write $f^t$ compactly as

$$f^t((H_v^{t-1}, H_c^{t-1})) = (H_v^t, H_c^t)$$

$$= \left( U_v^t \left( H_v^{t-1}, [A_+ \quad A_-] \begin{bmatrix} M_{v+}^t(H_c^{t-1}) \\ M_{v-}^t(H_c^{t-1}) \end{bmatrix} \right), U_c^t \left( H_c^{t-1}, \begin{bmatrix} A_+ \\ A_- \end{bmatrix}^\top \begin{bmatrix} M_{c+}^t(H_v^{t-1}) \\ M_{c-}^t(H_v^{t-1}) \end{bmatrix} \right) \right).$$

For the same graph, the policy network computes a probability vector $\hat{p} \in \mathbb{R}^n$ over variables as

$$\mathbf{H}^T = \left( f^T \circ f^{T-1} \circ \cdots \circ f^1 \right) (\mathbf{H}^0)$$

$$\hat{p} = \mathrm{softmax}\left( Z(H_v^T) \right)$$

where $\mathrm{softmax}(y) = \exp(y)/\sum \exp(y_i)$. In order to refer to the above computation concisely we define the function $\pi_\theta$ computed by the policy network as $\hat{p} = \pi_\theta(\phi, X)$ where we assume $\mathbf{H}^0$ will be obtained from the initial assignment $X$, and $\mathbf{A}$ will be obtained from the formula $\phi$. As before, $\theta$ is the list of all neural network parameters. In our implementation, all of $M_{v+}^t, M_{v-}^t, M_{c+}^t, M_{c-}^t, U_v^t, U_c^t, Z$ are multilayer perceptrons (MLP). Also, we use $T = 2$ which is the minimum number of iterations to allow messages to travel between variables that occur together in a clause. Remaining hyperparameters are described in Appendix A.

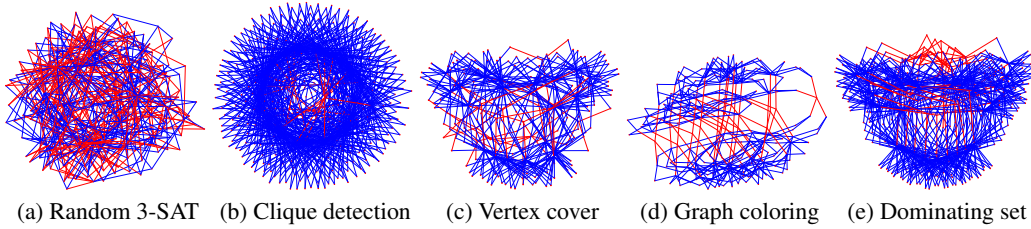

| (a) Random 3-SAT | (b) Clique detection | (c) Vertex cover | (d) Graph coloring | (e) Dominating set |

Figure 2: Examples of factor graphs exhibiting distinct structures, each corresponding to problems sampled from different distributions. Structures of the graphs corresponding to formulas have been studied in the SAT solving community as a way to explain the effectiveness of CDCL [6] on industrial problems [2]. With a similar intuition, our aim is to learn heuristics that can exploit the structure.

## 5   Data

As our goal is to learn specialized heuristics for different classes of satisfiability problems, we generate problems from various distributions to train on. There are five problem classes we perform experiments with: random 3-SAT, clique detection, vertex cover, graph coloring, dominating set. Table 1 presents the notation for the problem classes that we consider.

Table 1: As before, $n$ and $m$ refer to the number of variables and clauses. For graph problems $N$ refers to the number of vertices in the graph and $p$ to the probability that an edge exists. $k$ refers to the problem specific size parameter. From each distribution $D$ on the table we can sample a CNF formula $\phi \sim D$.

| Class | Distribution |
|---|---|
| Random 3-SAT | $\mathrm{rand}_3(n, m)$ |
| k-clique | $\mathrm{clique}_k(N, p)$ |
| k-cover | $\mathrm{cover}_k(N, p)$ |
| k-coloring | $\mathrm{color}_k(N, p)$ |
| k-domset | $\mathrm{domset}_k(N, p)$ |

Random 3-SAT[1] is of theoretical interest in computational complexity and also serves as a common benchmark for SLS-based SAT algorithms. The latter four are NP-complete graph problems. For each of these problems we sample Erdős–Rényi graphs [16] from the distribution denoted as $G(N, p)$ to mean that it has $N > 0$ vertices and between any two vertices an edge exists independently with probability $p \in [0, 1]$. Then we encode them as SAT problems. As a result we obtain five parameterized random distributions that we can sample formulas from. For training and evaluation we generate problems of various sizes (made explicit later) from these five families of distributions. It is worth noting that the generated problem instances are not particularly difficult for state-of-the-art SAT solvers, and for our purpose they serve as simple benchmarks to help demonstrate the feasibility of a purely learning-based approach.

In creating problem instances we use CNFgen [30] to generate formulas and Minisat [15] to filter out the unsatisfiable ones. Since our local search algorithm is an incomplete solver it can only find an assignment if one exists and otherwise returns unsolved.

## 6   Training

### 6.1   Markov decision process formulation

To learn heuristics through reinforcement learning [44], we formalize local search for SAT as a Markov decision process (MDP). For each problem distribution $D$ shown on Table 1 we have a separate MDP represented as a tuple $(\mathcal{S}_D, \mathcal{A}, \mathcal{T}, \mathcal{R}, \gamma)$ with the following components:

- $\mathcal{S}_D$ is the set of possible states. Each state is characterized by a pair $s = (\phi, X)$, the CNF formula and a truth assignment to its variables. At the start of an episode we sample a formula $\phi \sim D$ with $n$ variables and $m$ clauses, and a uniformly random initial assignment

$X \in \{0, 1\}^n$ where each element is either 0 or 1 with probability 1/2. The episode terminates either when we arrive at a satisfying assignment or after $L$ (a predetermined constant) steps are taken.

- $\mathcal{A}$ is formally a function that maps states to available actions. For a state $s = (\phi, X)$ we have $\mathcal{A}(s) = \{1, \ldots, n\}$ where $n$ is the number of variables in $\phi$.

- $\mathcal{T} : \mathcal{S}_D \times \{1, \ldots, n\} \to \mathcal{S}_D$ is the transition function, mapping from a state-action pair to the next state. It is defined as $\mathcal{T}(s, a) = \mathcal{T}((\phi, X), a) = (\phi, \mathsf{Flip}(X, a))$ where $\mathsf{Flip}$ simply negates the assignment of the variable indexed by $a \in \{1, \ldots, n\}$.

- $\mathcal{R} : \mathcal{S}_D \to \{0, 1\}$ is the reward function, defined as $\mathcal{R}(s) = \mathcal{R}((\phi, X)) = \phi(X)$, that is, 1 for a satisfying assignment and 0 otherwise.

- $\gamma \in (0, 1]$ is the discount factor, which we set to a constant strictly less than 1 in order to encourage finding solutions in fewer steps.

With the MDP defined as above, the problem of learning a good heuristic is equivalent to finding an optimal policy $\pi$ which maximizes the expected accumulated reward obtained when starting at a random initial state and sampling a trajectory by taking actions according to $\pi$. In trying to find an optimal policy, we use the REINFORCE algorithm [47]. As our policy we have a function $\rho_\theta(\phi, X)$ which returns an action (variable index) $a \sim \pi_\theta(\phi, X)$ where $\pi_\theta$ is the policy network we described in Section 4.3. With probability 1/2, $\rho_\theta$ returns a randomly selected variable from a randomly selected unsatisfied clause.

When training to learn a heuristic for a problem distribution $D$, at each training iteration we sample a formula $\phi \sim D$ and generate multiple trajectories for the same formula with several different initial assignments $X$. Then we accumulate the policy gradient estimates from all trajectories and perform a single update to the parameters. Algorithm 2 in Appendix B shows the pseudocode for training.

## 6.2 Curriculum learning

In our MDP formulation, a positive reward is achieved only when an episode ends at a state corresponding to a satisfying assignment. This means that to have non-zero policy gradient estimates, a solution to the SAT problem has to be found. For difficult problems, we are unlikely to arrive at a solution by taking uniformly random actions half the time, which is what happens at the beginning of training with a policy network that has randomly initialized parameters. This may not prevent learning, but makes it prohibitively slow. In order to solve this problem, we opt for curriculum learning [8]. With curriculum learning, training is performed on a sequence of problems of increasing difficulty. The intuition is that the policy learned for easier problems should at least partially generalize to more difficult problems, and that this should lead to faster improvement compared to training directly on the difficult problem of interest. Specifically, we follow an approach where we run the training loop in sequence for distributions of increasing size within the same problem class.

As an example, assuming our goal is to learn heuristics for $\mathrm{rand}_3(25, 106)$, we begin by training to solve $\mathrm{rand}_3(5, 21)$ for which a positive reward is obtained often enough to yield a quick improvement in the policy. This improvement translates to larger problems, and we continue training the same policy network to solve $\mathrm{rand}_3(10, 43)$ for which it becomes easier to quickly find a solution compared to starting from scratch. More specifically, at the $i$th curriculum step we train on a small distribution $D_i$ for a fixed number of steps while evaluating on a slightly larger distribution $D_{i+1}$. For the next step, we train on $D_{i+1}$ beginning with the model parameters from before that took the lowest median number of steps during evaluation on $D_{i+1}$. In this manner we keep stepping up to larger problems and finally we train on the distribution of the desired size.

## 7 Experiments

**Evaluation**    As a baseline we have the SLS algorithm WalkSAT as described by Selman et al. [39]. Note that although there have been improvements over WalkSAT in the last three decades, we selected it due to its simplicity and its foundational role in the development of the state-of-the-art SAT solvers that use SLS [3, 10]. That being said, competing with the state-of-the-art SAT solvers in run time is not our primary goal as the scalability of the heuristic computed by the GNN is currently bound to be lacking, which we expect will be alleviated by the ongoing rapid advancements in

Table 2: Performance of the learned heuristics and WalkSAT. In the first column, $n$ and $m$ on the second line of each cell refers to the maximum number of variables and clauses in the sampled formulas. For graph problems, the size of the graph $G(N, p)$ and the size of the factor graph corresponding to the SAT encoding are different. At each cell, there are three metrics (top to bottom): average number of steps, median number of steps, percentage solved.

| | $\text{rand}_3(n, m)$ | $\text{clique}_k(N, p)$ | $\text{cover}_k(N, p)$ | $\text{color}_k(N, p)$ | $\text{domset}_k(N, p)$ | WalkSAT |
|---|---|---|---|---|---|---|
| $\text{rand}_3(50, 213)$ | **367** **273** **84%** | 743 750 0% | 749 750 0% | 736 750 0% | 642 750 20% | 385 297 80% |
| $\text{clique}_3(20, 0.05)$ $n = 60, m = 1725$ | 529 750 48% | **116** **57** **100%** | 623 750 16% | 743 750 0% | 725 750 0% | 237 182 100% |
| $\text{cover}_5(9, 0.5)$ $n = 55, m = 790$ | 749 750 0% | 750 750 0% | **181** **115** **100%** | 750 750 0% | 224 162 100% | 319 280 96% |
| $\text{color}_5(20, 0.5)$ $n = 100, m = 480$ | 675 750 16% | 748 750 0% | 750 750 0% | **342** **223** **88%** | 645 750 16% | 416 379 80% |
| $\text{domset}_4(12, 0.2)$ $n = 60, m = 995$ | 729 750 0% | 660 750 16% | 304 169 76% | 748 750 0% | **205** **121** **100%** | 217 140 100% |

domain-specific processors. Hence, WalkSAT serves mostly as a "sanity-check" for the learned heuristics. This has been the case for other purely neural network-based approaches to SAT solving or combinatorial optimization, and these kinds of studies are currently more of scientific than practical interest. Nevertheless, we provide a comparison to WalkSAT.

In order to evaluate the algorithms, we sample 50 satisfiable formulas from five problem distributions to obtain evaluation sets, and perform 25 search trials (with each trial starting at a random initial assignment) using the learned heuristics and WalkSAT. Each algorithm runs for a maximum of 750 steps unless specified otherwise and has a walk probability of $1/2$. We model our evaluation methodology after that of KhudaBukhsh et al. [25] and report three metrics for each evaluation set on Table 2: the average number of steps, the median of the median number of steps (the inner median is over trials on each problem and the outer median is over the problems in the evaluation sets), the percentage of instances considered solved (median number of steps less than the allowed number of steps). For all the results reported in the next section we follow the same method. Our implementation is available at `https://github.com/emreyolcu/sat`.

### 7.1 Results

**Comparison to WalkSAT** Table 2 summarizes the performance of the learned heuristics and WalkSAT. Each column on the table corresponds to a heuristic trained to solve problems from a certain class. For each heuristic we follow a curriculum as explained in Section 6.2, that is, we train on incrementally larger problems of the same class and finally train on the distribution that we want to evaluate the performance on. Each row of the table corresponds to an evaluation set.

After training, the learned heuristics require fewer steps than WalkSAT to solve their respective problems. The difference is larger for graph problems than it is for random 3-SAT, which makes sense as there is no particularly regular structure to exploit in its factor graph. While the number of steps is reduced, we should emphasize that the running time of our algorithm is much longer than WalkSAT. With this work, our goal has not been to produce a practical SAT solver, but to demonstrate that this approach might allow us to create specialized algorithms from scratch for specific problems.

**Specialization to problem classes** As expected, the performance of each heuristic degrades on unseen classes of problems compared to the one it is specialized to. This provides evidence that the learned heuristics exploit class-specific features of the problems. Also, it is interesting to note that the performance of the learned heuristics on vertex cover and dominating set problems correlate. They are indeed similar problems, and it is not surprising to see that a good heuristic for one also performs well for the other. Their similarity is also visible from their example factor graphs in Figure 2.

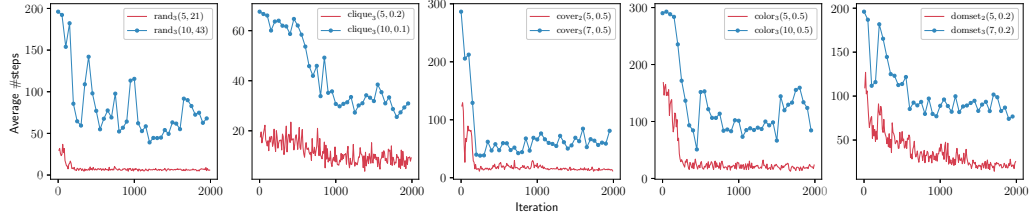

Figure 3: Curriculum during training. The plots display the average number of steps taken over multiple trials during training when learning to solve SAT problems. Each plot includes two curves, one for the small distribution from which we sample training problems, and another for a fixed set of evaluation problems from a larger distribution of the same problem class.

**Effect of the curriculum** Example training runs of the curriculum that we employ are shown in Figure 3. They demonstrate that the learned heuristic for a smaller distribution transfers to a slightly larger distribution from the same class, which allows us to step up to larger problems.

**Generic heuristics** Each learned heuristic is expected to exploit the specific structure of its problem class as much as possible. Intuition suggests that when there is no discernible structure to exploit, the resulting heuristic may generalize relatively well to other classes. As an experiment, we learn a heuristic to solve satisfiable formulas from the $\mathbf{SR}$ distribution, created by Selsam et al. [see 42, section 4] to generate formulas that are difficult to classify as satisfiable or unsatisfiable based on crude statistics. We then evaluate this heuristic on the graph problems. Random 3-SAT near the threshold is another similarly difficult distribution, although its difficulty is an asymptotic statement and not necessarily useful for our case with small formulas. Still, we repeat the experiment with random 3-SAT for completeness. Table 3 shows that a heuristic learned on $\mathbf{SR}(10)$ generalizes better to the slightly larger and different problem distributions compared to a heuristic learned on $\mathrm{rand}_3(10, 43)$.

Table 3: Generalization from $\mathbf{SR}(10)$ to different problem distributions. Each cell displays the average number of steps and the percentage of problems solved.

|  | $\mathbf{SR}(10)$ | $\mathrm{rand}_3(10, 43)$ |
|---|---|---|
| $\mathrm{clique}_3(10, 0.1)$ $n = 30, m = 420$ | 363 98% | 649 4% |
| $\mathrm{cover}_4(8, 0.5)$ $n = 40, m = 450$ | 410 86% | 643 16% |
| $\mathrm{color}_4(15, 0.5)$ $n = 60, m = 200$ | 204 98% | 390 84% |
| $\mathrm{domset}_4(12, 0.2)$ $n = 60, m = 995$ | 499 70% | 653 16% |

**Search behavior** Table 2 shows that the largest differences between the learned heuristics and WalkSAT are on clique detection and vertex cover problems. In order to gain an understanding of how the learned heuristics behave differently, we look at a few statistics of how they traverse the search space. Table 4 shows the ratio of the flips during the search that flip a previously flipped variable (undo), move to a solution that increases (upwards), does not change (sideways), or decreases (downwards) the number of unsatisfied clauses. The reported statistics are computed by taking into account only the flips that are made deterministically by the heuristics, although if a variable was previously randomly flipped and the heuristic chooses to flip the same variable again then this is counted as an "undo". The striking difference is that the learned heuristics make sideways moves far less often, and instead appear to zoom in on a solution with downwards moves. It is also intriguing that they make "bad" moves (upwards) with relatively high frequency.

Table 4: Macroscopic comparison of the search behaviors of the learned heuristics and WalkSAT.

| Heuristic | $\mathrm{clique}_3(20, 0.05)$ | | | | $\mathrm{cover}_5(9, 0.5)$ | | | |
|---|---|---|---|---|---|---|---|---|
|  | Undo | Upwards | Sideways | Downwards | Undo | Upwards | Sideways | Downwards |
| Learned | 0.26 | 0.28 | 0.14 | 0.57 | 0.37 | 0.38 | 0.13 | 0.48 |
| WalkSAT | 0.33 | 0.16 | 0.54 | 0.29 | 0.39 | 0.24 | 0.40 | 0.35 |

**Different random graphs**  Above experiments on SAT encoded graph problems all use the Erdős–Rényi model for random graphs. Although the results on Table 2 provide evidence that each heuristic is specialized to its respective problem class, they do not allow us to conclude that the heuristics should still work for different random graphs. Ideally, the heuristic learned on graph coloring problems should more heavily be exploiting the fact that the formula is the SAT encoding of a coloring problem than the fact that the encoding is for a graph sampled according to the Erdős–Rényi model.

To test whether the heuristics can generalize to different random graphs, for each graph problem we generate satisfiable instances on graphs from four distributions (random regular [26], random geometric [36], Barabási–Albert [4], Watts–Strogatz [46]) with the number of vertices varying from 10 to 15 and use these instances to re-evaluate the previously learned

Table 5: Performance of the learned heuristics and WalkSAT on problems with graphs from distributions unseen during training. Each cell displays the average number of steps and the percentage of problems solved.

|  | Learned | WalkSAT |
|---|---|---|
| $\text{clique}_k$ | 198 | 285 |
| $3 \le k \le 5$ | 100% | 98% |
| $\text{cover}_k$ | 306 | 370 |
| $4 \le k \le 6$ | 90% | 92% |
| $\text{color}_k$ | 162 | 193 |
| $3 \le k \le 5$ | 100% | 100% |
| $\text{domset}_k$ | 271 | 255 |
| $2 \le k \le 4$ | 92% | 92% |

heuristics. Table 5 shows the results compared to WalkSAT. Each row on the table corresponds to a set of 60 problems with varying problem specific size parameters $k$ and graphs sampled from the four distributions. While the shift in the evaluation distribution causes some decline in the relative performance of the learned heuristics against WalkSAT, they can still find solutions in fewer steps than WalkSAT most of the time. Recall that the goal here is not necessarily to perform better than WalkSAT, but being able to do so provides evidence of the robustness of the learned heuristics.

**Larger problems**  In order to show the extent to which a learned heuristic can generalize to larger problems, we run the heuristic learned only on the small problems from $\mathbf{SR}(10)$ to solve satisfiable problems sampled from $\mathbf{SR}(n)$ for much larger $n$ with a varying number of maximum steps. Figure 4 shows that as we increase the number of maximum iterations, the percentage of problems solved scales accordingly with no sign of plateauing. For comparison, WalkSAT results are also included.

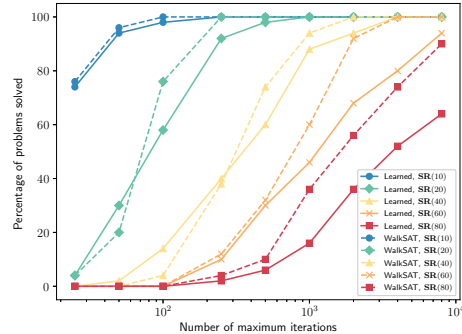

Figure 4: Generalization of the $\mathbf{SR}(10)$ heuristic to larger problems. Solid and dashed lines correspond to the learned heuristic and WalkSAT, respectively.

# 8   Conclusion

We have shown that it is possible to use a graph neural network to learn distribution-specific variable selection heuristics for local search from scratch. As formulated, the approach does not require access to the solutions of the training problems, and this makes it viable to automatically learn specialized heuristics for a variety of problem distributions. While the learned heuristics are not competitive with the state-of-the-art SAT solvers in run time, after specialization they can find solutions consistently in fewer steps than a generic heuristic.

There are a number of avenues for improvement. Arguably, the most crucial next step would be to reduce the cost of the heuristic to scale to larger problem instances. If a GNN is to be used for computing the heuristic, more lightweight architectures than we have can be of interest. Also, in this work we focused on stochastic local search, however, the SAT solvers used in practice perform backtracking search. Consequently, model-based algorithms such as Monte Carlo tree search [12] can provide critical improvements. As a step towards practicality, it is also important to incorporate the suite of heuristics in an algorithm portfolio. In this work, we have achieved promising results for learning heuristics, and we hope that this helps pave the way for automated algorithm design.

## Footnotes

[1]For random 3-SAT, experimental research [40, 14] indicates that formulas with a ratio of the number of clauses to the number of variables approximately $4.26$ and a large enough number of variables are near the satisfiability threshold, that is, the probability of a sampled formula being satisfiable is near $1/2$. We focus on random 3-SAT problems near the threshold.

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
