[Supplementary Material]

# Appendix

## A    Hyperparameters

We tune almost all hyperparameters for random 3-SAT and use the same ones for the other problem classes. For optimization we use RMSprop [45]. At the first curriculum step (smallest distribution), we start with a learning rate of 0.001 and halve it at predefined points in training which continues for 10000 iterations. For subsequent curriculum steps we start with a learning rate of 0.0001 and multiply it by 0.8 at predefined points. During training we sample 5 trajectories, and set the maximum allowed number of steps per distribution, determined by the maximum number of steps that WalkSAT takes over a sample of problems from the same distribution. We use a discount factor of $\gamma = 0.25$. We realize this is an unconventional choice for the discount factor. However, in our experiments, using a small discount translated to faster convergence, especially in the initial curriculum steps.

Each of the functions described in Section 4.3 are MLPs with a single hidden layer and ReLU activation (except for the final layer of $Z$, after which we apply softmax). The GNN runs $T = 2$ message-passing iterations, and after each iteration we perform batch normalization [23] where we treat all the nodes in the factor graph as a minibatch. We observed that batch normalization was crucial to achieve our results. In experiments using $T > 2$ we saw no benefits to performance. This also goes along with the conventional wisdom that the intuition from supervised learning with neural networks where larger and deeper models are often better might not be applicable to reinforcement learning. For all message and update MLPs we have $d = 32$, and the hidden layer also has size 32 for all MLPs except $Z$, which has 64 hidden units.

We also experimented with an actor-critic architecture and entropy regularization. For the set of hyperparameters that we searched over, these more complex approaches were on par with or worse than vanilla REINFORCE.

## B    Training algorithm

---
**Algorithm 2** Learning a heuristic with REINFORCE
---
**Input:** Problem distribution $D$, policy $\rho_\theta$, discount rate $\gamma$, learning rate $\alpha$
 1: Initialize parameters of the policy $\rho_\theta$
 2: **for** $i \leftarrow 1$ to NumTrainIterations **do**
 3:     Sample a problem $\phi \sim D$
 4:     Initialize $g \leftarrow \mathbf{0}$                                              ▷ $g$ accumulates gradients.
 5:     **for** $j \leftarrow 1$ to NumTrajectories **do**
 6:         Initialize history $= ()$
 7:         Assign $X \leftarrow$ Initialize$(\phi)$
 8:         **for** $k \leftarrow 1$ to MaxFlips **do**
 9:             **if** $\phi(X) = 1$ **then**                                        ▷ $X$ satisfies $\phi$.
10:                 **break**
11:             **else**
12:                 Sample action $a \sim \rho_\theta(\phi, X)$
13:                 Append $(X, a)$ to history
14:                 $X \leftarrow$ Flip$(X, a)$
15:             **end if**
16:         **end for**
17:         Set reward $r \leftarrow \phi(X)$
18:         **for** $t \leftarrow 1$ to $T = $ length(history) **do**
19:             $X, a \leftarrow$ history$[t]$
20:             **if** $a$ was sampled using $\pi_\theta$ **then**
21:                 $\hat{p} \leftarrow \pi_\theta(\phi, X)$
22:                 $g \leftarrow g + \gamma^{T-t} r \, \nabla_\theta \log \hat{p}(a)$
23:             **end if**
24:         **end for**
25:     **end for**
26:     $\theta \leftarrow \theta + \alpha g$                                             ▷ Update policy parameters.
27: **end for**
---