[Reviews · NeurIPS 2019]

Reviewer 1



Originality People have tried local search for some combinatorial optimization problems before, but I could not find any references to boolean satisfiability in particular. Doing curriculum training when there is a measure of difficulty, is not new. Clarity I found it clear. Quality The authors do the first step in replacing CDCL SAT Solvers, by having a model that decides which variable to flip next. The findings support the claims, although the evaluation could have been on slightly more difficult instances (e.g. that WalkSAT doesn't solve) Significance The value of SAT Solving is finding solutions to problem in the real world (plus human modelling), industrial instances were not evaluated or trained on, so it is diffcult for me to measure this.

Reviewer 2



1) Ablation studies. It would be really interesting which algorithmic choices were necessary to get the reported results eg: What about longer unrolls of the graph net T>2? How important is it to hard code that the policy with p=1/2 flips random variables of unsatisfied clauses? 2) Details on curriculum. I did not find the details to what criterion was used to decide when to ratchet up the task difficulty in curriculum learning. Could the authors clarify (or did I miss it somewhere)? 3) No value function? I’m surprised to see that the authors used straight-up REINFORCE without even a value-baseline (as explained in the appendix), bc in my experience this almost always makes a big difference in learning speed. Have the authors tried just adding a different “head” to the graph net torso that predicts expected return and use that as a baseline? 4) Figure 4: It would be interesting to see the walk-SAT baseline results here in the graph. 5) l228-231: Why is the policy trained on multiple episode on the same problem? Is this to reduce computational overhead due to initial checking for satisfiability with MiniSAT? I’m asking because attempting to solve the same problem multiple times can cause strong correlation in policy gradient. It might give a significant boost to write feasible instances into a replay buffer and then sample random formulas afresh for each gradient update.

Reviewer 3



This work is original in its use of deep reinforcement learning and graph neural networks to learn novel search control heuristics for SAT solving. While the techniques used are not novel themselves, the application domain is. The authors do a good job of surveying related work in this area and situating their contributions in this landscape. The paper is well-written and I found it very easy to follow the details of the proposed approach and the authors' results. Technically, the work presented is solid, though I have a few comments/suggestions here. Firstly, while I take the authors' point about not trying to compete on solution time with state-of-the-art SLS solvers that have been engineered and fine-tuned for decades, I think it would be worth comparing the performance of their algorithm to that of survey propagation (SP). SP, besides being one of the strongest SLS solvers on random 3-SAT, actually uses a message-passing approach as well -- and thus would make for a particularly interesting comparison with the authors' GNN-based algorithm. I laud the authors' direct and honest consideration of the strengths and drawbacks of their approach. I appreciate that they embrace the scientific interest of this work (over simply trying to advance the state-of-the-art in solver performance). I don't think they should be coy about reporting other things that may be of interest to the broader community: for example, the solution time, the gap to SP etc. Such performance numbers, even if they compare unfavorably to WalkSAT or SP, would give the community an idea of what the current gap is, and a way to measure future progress in this direction. Generalization from rand_3 (lines 287--297): interestingly, my intuition reading this section was the exact opposite! My feeling is that a heuristic that works well in formulas with no structure would *not* translate well to domains with structure -- indeed, the SAT contests traditionally have three "tracks" for this reason: random instances, "crafted" instances (like the graph problems), and industrial instances that come from applications. I would consider reframing this discussion. Table 4 raises an intriguing question: how does this divide in performance between WalkSAT and the GNN approach grow as the problem size increases? For example, it may be that the GNN+SLS combination zooms in faster on the solution, so you might even beat WalkSAT on time for large enough instances, where WalkSAT's meandering may lead to long runtimes. Finally, a couple of minor comments: - The caption for figure 3 could be clearer (which curve is which?) - Table 4: are we only looking at flips that are recommended by the heuristic, or are random flips included as well?

[Author Response · NeurIPS 2019]

We thank all the reviewers for their comments and criticisms. In our response we will answer the specific questions
raised and attempt to clear up a few points. In the three sections that follow we respond to the comments in each review.

**Response to R1.** 1. (Local search for SAT reference) In the paper, we reference WalkSAT as the basic local search
algorithm for Boolean satisfiability. Also, in Algorithm 1 we give a generic pseudocode of local search for SAT.
Proceedings of the recent SAT conferences and SAT competitions include SLS-based solvers more sophisticated than
WalkSAT, however most if not all of them already fit in to the pseudocode that we provided.
2. Doing curriculum learning when there is a measure of difficulty is indeed not new, and we use it solely to accelerate
training. As Reviewer 3 observes, the main novelty of this work is in the application domain.
3. R1 writes: "The authors do the first step in replacing CDCL SAT Solvers". This is slightly inaccurate in two ways:
We are not comparing at all against CDCL solvers, but rather the algorithms that perform stochastic local search (SLS),
in fact our approach can be thought of as an SLS algorithm template where the variable selection heuristic is learned
per problem class. Also, we would not yet phrase the goal as replacing CDCL solvers, as they are likely to stay popular
at least until the domain-specific hardware such as TPUs becomes common enough for approaches such as ours to
become significantly faster. Our focus lies in learning specialized heuristics from scratch with minimal supervision.
4. As for evaluation on more difficult instances, those not solved by WalkSAT are out-of-reach for our approach as
well. While the practical value of SAT solving is in solving real-world problems, investigations into entirely different
ways of approaching the problem holds scientific value, which we pursue with this work. Our current goal is not to beat
the state-of-the-art SAT solvers on industrial instances, but to show that a purely learning-based approach can help us
create algorithms with fine-grained specialization. Nevertheless, we are optimistic that a similar approach will hold
more practical value in the future.

**Response to R2.** 1. Longer unrolls were not beneficial enough (and sometimes even hurtful) to justify the increased
runtime. $p = 1/2$ is mostly arbitrary and not very important for the conclusions we draw. In our final revision we will
add ablation studies to the appendix performed on one of the problem distributions.
2. We have an additional explanation in Appendix C, although it does not answer your question. The specific way we
do curriculum is to train on a small problem distribution (say D1) for a fixed number of steps while evaluating on a
slightly larger distribution (D2). For the next step, we train on D2 beginning with the best model parameters from
before (lowest median number of steps on D2) while evaluating on a larger distribution and so on.
3. We did try using a different head as well as an entirely separate network to predict the value baseline. The MDP that
we defined is unorthodox in several ways, and many of our intuitions from deep RL actually seemed to fail. We will
release our code to allow people to experiment on their own and verify these.
4. We will include WalkSAT results on Figure 4 in our final revision.
5. Limiting the number of formulas seen during training actually accelerated training (with respect to evaluation
performance on larger, unseen problems from the same class). We did experiment with never reusing formulas
(checking satisfiability has a relatively low cost for the small problems that we use), which had an adverse effect.
Note that even when we recycle problems, the path taken (how the assignment changes) is often wildly different. Our
hypothesis is that not reusing the formulas makes the "environment" change too much between episodes (since the state
space is already large due to the number of possible assignments), hindering learning.

**Response to R3.** 1. At a first look, survey propagation (SP) may be worth comparing against, but SP on its own is not
particularly strong in solving formulas that are not uniform random k-SAT. Indeed, we performed this experiment at the
beginning of this work and SP (without decimation) did not converge to a solution on any of the SAT-encoded graph
problems. As a result, it is a baseline that is almost too weak for our purposes of sanity-checking the learned heuristic,
which is why we went with WalkSAT instead. As for the solution time compared against WalkSAT, it definitely will
be of interest to the broader community, but we do not want to be misleading since we did not focus too much on
the efficiency of our implementation (especially the graph neural network). Even with current hardware, once the
implementation is properly optimized (moving from Python to C as the first step), the real gap in achievable runtime is
probably lower than what we will report.
2. (Generalization from rand3) The learned heuristic is expected to exploit the specific structure of the problem class
as much as possible. When there is no structure to exploit, we hypothesized that the learned heuristic is generic,
because there is no way to "cheat" by exploiting the structure. WalkSAT, for instance, works well for the uniform
random formulas with no structure but is also able to solve some problems that exhibit structure (also see YalSAT and
probSAT for other examples of this). This is not a formal enough statement, but we think that the relationship between
DPLL–SLS is not the same as [learned heuristic on structured]–[learned heuristic on random]. We will reframe this
discussion since it seems to lead to confusion.
3. In Figure 3, the red curves correspond to the smaller distributions used for training and the blue curves correspond to
the larger ones used for evaluation. For Table 4, the reported numbers are computed by looking only at the flips that are
recommended by the heuristics (although if a variable was previously randomly flipped and the heuristic chooses to flip
the same variable again then this is counted as an "undo").

[Meta-Review · NeurIPS 2019]

The reviewers were positive about this paper based upon their initial read. The authors response addressed their concerns, so they were even more comfortable with a positive outcome after the author response. I encourage the authors to incorporate their responses to the reviewer concerns into any final version of the paper.